# Coverage and k-Coverage Optimization in Wireless Sensor Networks Using Computational Intelligence Methods: A Comparative Study

**Konstantinos Tarnaris [1], Ioanna Preka [2], Dionisis Kandris [1],\* and Alex Alexandridis [2]**

[1]  microSENSES Research Laboratory, Department of Electrical and Electronic Engineering, Faculty of Engineering, University of West Attica, GR-12241 Athens, Greece; ktarnaris@uniwa.gr

[2]  TelSiP Research Laboratory, Department of Electrical and Electronic Engineering, Faculty of Engineering, University of West Attica, GR-12241 Athens, Greece; ipreka@uniwa.gr (I.P.); alexx@uniwa.gr (A.A.)

\*  Correspondence: dkandris@uniwa.gr

**Abstract:** The domain of wireless sensor networks is considered to be among the most significant scientific regions thanks to the numerous benefits that their usage provides. The optimization of the performance of wireless sensor networks in terms of area coverage is a critical issue for the successful operation of every wireless sensor network. This article pursues the maximization of area coverage and area k-coverage by using computational intelligence algorithms, i.e., a genetic algorithm and a particle swarm optimization algorithm. Their performance was evaluated via comparative simulation tests, made not only against each other but also against two other well-known algorithms. This appraisal was made using statistical testing. The test results, that proved the efficacy of the algorithms proposed, were analyzed and concluding remarks were drawn.

**Keywords:** wireless sensor networks; coverage; k-coverage; optimization; computational intelligence; genetic algorithms; particle swarm optimization

## 1. Introduction

A wireless sensor network (WSN) is a network of wirelessly interconnected devices that are spread over an area of interest. The constituting elements of a WSN are the sensor nodes and at least one sink node, called the base station. A typical sensor node is a micro electromechanical system (MEMS) that, despite its constrained resources in terms of power, memory, communication, and computation, is able to perform sensing, data processing, and communicating tasks. A base station is device rich in energy, computational, and communication resources that collects the data transmitted to itself, by the sensor nodes, and acts as a supervisory controller of the WSN, an access point for human interface, and a gateway to other networks [1,2]. A WSN, based on the combinational use of its sensor nodes and its base station(s), is able to monitor the ambient conditions over wide regions of interest and send relevant information to distant destinations. This is why WSNs are considered to be the core of Internet of Everything (IoE) and support an endlessly evolving range of human activities related with industry, agriculture, surveillance and reconnaissance, smart homes, smart cities, environment and habitat monitoring, biomedical applications, military applications traffic control, fire detection, inventory control, agriculture, machine failure diagnosis, and energy management applications [3–10].

On the other hand, the aforementioned limitations of the sensor nodes in terms of their resources, and the inborn constraints of wireless communication channels, regarding power, speed, robustness, and capacity, obstruct the operation of WSNs and raise several issues that need to be addressed.

Among them, a major subject of research is the coverage problem. It can be stated as the problem of positioning a given number of sensor nodes with a given range of sensing in an optimal pattern in

order to maximize the proportion of the area covered by them and thus minimize the holes of coverage. Various methodologies for coverage maximization have been proposed [11–15]. In the research work presented in this article, the solution to this problem has been pursued based on two Computational Intelligence algorithms. The efficacy of these two algorithms has been investigated in comparison not only against each other but also contrary to two other well-known algorithms.

In this context, the rest of this article is organized as follows. In Section 2, the theoretical background regarding the topic under investigation is established. In Section 3, the algorithms developed are described. Both the simulation procedure developed and the corresponding results produced are presented in Section 4. In Section 5, the performance of the algorithms developed is evaluated and discussed. Finally, in Section 6, concluding remarks are drawn and future research topics related to the area coverage problem are identified.

## 2. Theoretical Background

### 2.1. Coverage in WSNs

In accordance to the subject of interest, three types of coverage in WSNs may be identified: area coverage, point coverage, and barrier coverage. Area (or regional) coverage expresses the ability of the network to monitor an area of interest, meaning that all points within this area are always monitored. Point (or target) coverage refers to the ability of the network to guarantee that a predetermined group of points are always observed. Barrier (or path) coverage refers to the ability to always detect the movement across a barrier of sensor nodes [11]. Actually, area coverage is the type of coverage problem that is mostly investigated in WSNs. It may be referred either as 1-coverage or as k-coverage (where k is an integer number greater than 1), depending on the minimum number of sensor nodes that must concurrently sense the area of interest.

It is obvious that for every WSN consisting of a specific number of sensor nodes, the maximization of the area coverage is desired. However, there are various factors that affect area coverage. For example, the deployment of sensor nodes can be either random or deterministic. Likewise, the sensitivity of sensor nodes may be either Boolean or probabilistic. Also, the sensing area may be deterministic or probabilistic. Similarly, the communication range of sensor nodes may be invariable or variable. Additionally, sensor nodes may be static or mobile. Moreover, the coverage scheme adopted may be either centralized or distributed [12]. For all these reasons, the area coverage problem is considered to be a non-trivial problem.

### 2.2. Computational Intelligence Methods

Computational intelligence (CI) is the scientific domain that uses nature-inspired computational methodologies in order to cope with problems for which conventional mathematical reasoning and modelling can be inadequate, due to the complexity, the uncertainty, or the stochastic nature of them [16]. Genetic algorithms and the particle swarm optimization algorithm are considered to be among the CI methods that are most widely used for optimization [17–22].

#### 2.2.1. Genetic Algorithms

Genetic algorithms (GAs) are a category of algorithms that search to find the optimal solution, i.e., a solution that maximizes or minimizes a particular function, to given computational problems. In order to do so, they imitate the operation of evolution. A population consisting of possible solutions is built, and new solutions are generated by breeding the best solutions found within all the members of the population in order to create a new generation. The population develops for many generations so that when the execution of the algorithm is completed the best solution is found [23].

Specifically, in GAs every individual represents a possible solution to an optimization problem. The characteristics of each individual are described by the chromosomes (also known as genomes) and constitute the variables of this problem. Before executing a GA, the chromosomes should be suitably encoded, in order to reflect the given problem. Additionally, a proper fitness function should be defined, in order to describe the problem as well as attribute a certain degree of fitness to each

encoded solution. During the execution of the genetic algorithm, the best parents for crossover and reproduction of descendants are selected. It is of outmost importance that these kinds of algorithms are constructed in such a way so that the diversity among the individuals of the population is ensured. In this way, the premature convergence of the algorithm can be prevented. One of the operators which strengthens the diversity of the genetic algorithm—and, thus, prevents its premature convergence—is the mutation operator; through its application, the random appearance of a chromosome could be noticed, which otherwise would be impossible to be detected through the crossover. In order to balance the size of the population in the new generation, some of the parents in the produced population are relocated, so its total size remains intact [24].

### 2.2.2. Particle Swarm Optimization

The Particle swarm optimization (PSO) algorithm is another CI method, inspired by nature, used for the solution of problems of various types. Specifically, PSO proposes the existence of a group of particles, where each of them represents a potential solution to the problem. Actually, the integral part of collectivity is the cooperation among the members that comprise it. The particles that create a specific collectivity interact with each other in order to achieve their common aim by exchanging information. In this way, each problem that arises can be solved in a more efficient way compared to what a sole particle can do. In nature, various kinds of such collectivities have been observed, many of which are organized in swarms. The main idea for the creation of PSO was based on the study of the synchronized flying patterns of birds and their ability to regroup appropriately when being members of flocks [25].

The notion of the swarm could be compared to the notion of population in genetic algorithms, whereas the notion of the particle could be linked to the one of the chromosome. The PSO particles work together in order to mimic the success of their neighboring particles, aiming at unveiling the best positions within the search space. In this algorithm the particles "fly" in the search space, thus, occupying certain positions which occur from the experience that has been gained from their own "flying" as well as from the one of their neighbors. Specifically, a certain number of particles is randomly placed in the search space and the result of the fitness function for the given position is assessed. Afterwards, for a predefined number of executions of the algorithm, each particle occupies a new position for which the fitness function gives a better result compared to the previous one. Each particle holds a memory of its best position so far; this is compared and contrasted with the best positions which have been occupied by the neighboring particles of the swarm and the best one is recorded. Therefore, as the number of executions of the algorithm is increased, the swarm achieves the fittest solution to the problem, having a predefined number of particles working for the accomplishment of their common goal. The search for the best solution through the PSO algorithm is an ongoing process which is executed continuously until the point a certain termination criterion is satisfied [26].

### 2.3. Strategies for Coverage Optimization in WSNs

As explained in Section 2.1 there are different types of coverage and factors that affect them. In order to cope with their particularities, various strategies have been proposed regarding the coverage problem in WSNs. The most popular of them are: grid based, virtual force based, and computational geometry based.

Grid based methods consider grid points in order to determine the location of sensor nodes and calculate the existent area coverage as the ratio of grid points covered to the total number of grid points. The commonly used grids are of triangular, hexagonal, square, and honeycomb types.

Force based strategy supposes the existence of virtual attractive and repulsive forces, which coerce sensor nodes to move towards or away each other until evenness is achieved so that full coverage is achieved.

In computational geometry based strategy the area to be covered is divided into geometrical structures, such as Voronoi diagrams and Delaunay triangulation. A Voronoi diagram is a polygon,

which is created by the lines of perpendicular bisectors that connect two neighboring objects. A Delaunay triangulation is a triangulation of an area such that no points in any triangle are located within the circumscribed circle of any other triangle in the area. Delaunay triangulation is the dual of Voronoi diagrams, because the circumcenters of Delaunay triangles are the vertices of the Voronoi diagram [27,28].

### 2.4. Related Work

In [29] the area coverage maximization is pursued through the appropriate positioning of mobile sensor nodes among stationary sensor nodes. The overall network field area is partitioned into a number of individual grids. At the same time, the weight of each individual grid is estimated. The target grid of any mobile node is considered to be the one that owns the minimum value of weight among all grids. In [30] WSN coverage optimization is pursued by using the PSO method along with a grid based strategy. Particularly, the square grid configuration is used to evaluate the particles in the fitness functions used. Additionally, maximizing the coverage and the minimization of energy consumption are also chased. The authors in [31] proposed a novel algorithm in order to optimize the coverage in a WSN using the PSO algorithm along with Voronoi diagrams. The points of interest are defined by the Voronoi diagram as the points at the edges of the Voronoi polygons as well as points in the boundaries where there is a high possibility that a coverage hole exists. According to the distance of the points of interest from the sensors, the area of coverage holes are computed and minimized by using a PSO algorithm.

In [32], the authors discovered that the phenotype space of the maximum coverage deployment problem in WSNs is a quotient space of the genotype space and proposed an efficient genetic algorithm using a novel normalization method. Additionally, they utilized the Monte Carlo method in order to design an efficient evaluation function, and achieved short computation times via a technique that starts from a few random samples and progressively raises their number for succeeding generations. In [33], a novel bidding protocol, which is applied in an area where a mixture of static and mobile nodes has been deployed, is proposed. According to the specific protocol, the mobile nodes change their locations in order to improve the coverage area by fulfilling the holes that have been determined by the static ones through the calculation of Voronoi diagrams.

A virtual force algorithm (VFA) is proposed in [34], as a sensor deployment methodology in order to improve the coverage area. The sensor nodes are initially deployed in a random way. Attractive and repulsive virtual forces are applied in the nodes in order to obtain a new position and once this position is determined they proceed to a one-time movement to their new optimal locations.

The authors in [35] proposed a virtual force-directed particle swarm optimization (VFPSO) algorithm. VFPSO is a self-organizing algorithm, aiming to enhance the coverage in WSNs that consist of both mobile and stationary node. The proposed algorithm combines the virtual force (VF) algorithm with particle swarm optimization, where VF uses both attractive and repulsive forces to determine virtual motion paths and the rate of movement for sensors. In this way, the velocities of particles are updated according to not only the historical local and global optimal solutions but also the aforementioned virtual forces of sensor nodes.

In [36] the so called coverage enhancing (COVEN) algorithm was introduced. COVEN calculates the percentage of hole area after random deployment of sensor nodes. Specifically, by using a two-step deployment process, static nodes cooperate in order to define the number of mobile nodes that need to be added in the sensor network, as well as their final location, so as to achieve maximization of coverage. In this way, a tradeoff between the cost of deployment and percentage of area covered is attained.

## 3. Proposed Algorithms

As already mentioned above, in the research work presented in this article two different optimization algorithms were developed, i.e., a GA and a PSO algorithm. Basic prerequisite for the proper function

of each of these two algorithms is the knowledge of the characteristics of the search space, where the wireless sensor network will be placed, as well as the population and the behavior of the sensor nodes which comprise it. It is assumed that the search space is squared, two-dimensional, and without any obstacles. Therefore, the sensor nodes are not affected as far as the position they can occupy is concerned, since they remain within the search space limits, and the sensing range is not affected by extrinsic factors. A significant parameter that has an effect on the final positions of the sensor nodes is the existence of points of interest (POI) which require k- coverage. The sensor nodes' function is based on the binary-disc sensing model, according to which they have 100% detection ability within the sensing range. The algorithms that have been designed entail two basic aims:

- The maximization of the coverage area with the use of a certain number of sensor nodes.
- The attainment of k-coverage in predefined points in the area.

Basic points towards finding the solution to such kind of problems with the use of CI methods are the input information encoding and the creation of an apt function that describes the problem.

### 3.1. Input Information Encoding

One of the basic points during the modeling process of a natural problem is the way with which the factors which affect it will be encoded. What is more, one of the basics prerequisites is the possibility for the user to control some of the parameters so as to form the case to be solved.

As far as the genetic algorithm developed is concerned, in every chromosome a possible solution to the problem of coverage and k-coverage in encoded, while the final solution of the problem represents the optimal positions where the nodes should be placed. The position of each node in the two-dimensional space is defined by a pair of coordinates (x, y) and if the number of the sensor nodes is M, then M pairs of coordinates are required to specify their positions. Also, if there are N points that require k-coverage, then N triad of values is needed, where every triad is composed by the coordinates of the point in the area and the k number which defines the necessary coverage for the point. All the aforementioned variables along with the number of nodes, the space limits, the sensing ranges of the nodes, and the sampling steps that are necessary in order to calculate the algorithm constitute the information which is encoded in every chromosome in the genetic algorithm. Respectively, as far as the PSO based algorithm is concerned, the same information is encoded in the particles of the swarm. The velocity and the position of each particle of the swarm were calculated according to Equations (1) and (2):

$$V_{ij}(t+1) = w * V_{ij} + c_1 * r_1 * \left( y_{ij}(t) - x_{ij}(t) \right) + c_2 * r_2 * \left( \hat{y}_j(t) - x_{ij}(t) \right) \tag{1}$$

$$x_i(t+1) = x_i(t) + V_i(t+1) \tag{2}$$

where $V_{ij}$ and $x_i$ represent the velocity and the position of the i-th particle (from the total n particles) in the $j$ dimension (from the $J$ total dimensions) during the t-th execution of the algorithm respectively. Additionally, w represents the inertia of the particle, $c_1$ and $c_2$ symbolize acceleration coefficients, and $r_1$ and $r_2$ are random numbers in the range [0, 1].

### 3.2. Fitness Function

The fitness function evaluates the quality of the solution being encoded in the chromosome or in the particle. The problem which is calculated by the algorithm is a minimization one, where the goal is to minimize the coverage holes, assuming that the k-coverage requirement is satisfied. Initially, the encoded information is decoded, making it possible for the function to process the design variables of the problem. The algorithm is comprised of two fundamental functions, the one that calculates k-coverage and the one that calculates the area coverage.

### 3.2.1. K-coverage Calculation

During the execution of the function, the number of points which require k-coverage as well as the degree of coverage are checked. For every spot, the algorithm withholds the necessary number of nodes which is equal to the degree of coverage of that spot; at the same time it keeps in its memory the nodes which have been already used for the coverage of previous spots. Each time, the Euclidian distance between the spot and the engaged group of sensor nodes to cover this spot is calculated. If the required coverage is satisfied, then the distance which occurs after the calculation is equal to zero. In every other case, the sum of the distances of the nodes from the spot is calculated. This process is repeated for all the spots which require k-coverage. The return value is the algebraic sum of all the distances that have been calculated from all the spots. It becomes apparent that if all the spots are covered, then the return value of the function is equal to zero. The methodology upon which the algorithm was developed is presented in Figure 1:

```
KcoverageCalculation:
   summation = 0
   distance = 0
   for(Every POI that demands k-coverage)do
      for(K equal to the required k-coverage of the POI)do
         d = Calculation of the distance between the node and the POI
            if (The selected node doesn't cover the target)
               distance = distance + d
            end
      end
   end
   return distance
end
```

**Figure 1.** K-coverage calculation algorithm.

### 3.2.2. Area Coverage calculation

The calculation of area coverage is based upon the sampling of the target area. More specifically, the more spots which are covered by the sensor nodes, the larger the coverage of the area. Initially, for every spot, it is examined whether at least one of the available nodes covers it. If so, the area coverage is increased by one, otherwise the coverage hole is increased in the same manner. At the point where all of the spots of the target area have been tested, the return value of the algorithm is the area coverage. The basic structure of the algorithm is presented in Figure 2.

```
CoverageCalculation:
   coverage = 0
   for(every spot of the squared area)do
      for(every node that it is placed in the field)do
         if(The node covers the spot)
            coverage = coverage + 1
         end
      end
   end
   return coverage
end
```

**Figure 2.** Area coverage calculation algorithm.

### 3.2.3. Fitness Function Calculation

The first step towards the execution of the fitness function is the calculation of the k-coverage of the spots which have been predefined by the user. In case its output is different than zero, the algorithm is terminated and its return value is equal the one that has been already calculated. Only in the case where the output is equal to zero can the algorithm proceed to the coverage area calculation. This methodology has been chosen since its goals are satisfied in a hierarchical manner; it reassures that a solution for the k-coverage for the spots will be found and, assuming it exists, it will proceed to find the solution that minimizes the holes' coverage. Moreover, in this way, the execution time is decreased, as the calculation of the coverage area, which is a time consuming process, is not executed unless the requirement of k-coverage is satisfied.

Through the execution of both the genetic and the PSO algorithms, the aim was to minimize the output of the fitness function. Thus, cases in which the k-coverage of the spots is not satisfied or the coverage area is not suitable related to the case are not considered optimal solutions, resulting in their rejection.

### 4. Simulation Tests and Results

In this section, the efficacy of both the genetic algorithm and the PSO algorithms in maximizing the coverage and the k-coverage of area in a WSN deployed at a two-dimensional squared area is investigated via analytical simulation tests performed in MathWorks MATLAB environment.

Specifically, seven case studies were studied. In the first four of them, the two aforementioned algorithms were compared against each other. In the last three case studies, the efficacy of these two algorithms was compared to that of a relevant grid based PSO algorithm proposed in [30] and a Voronoi-based PSO algorithm proposed in [31]. It must be noted that in order to ensure a fair comparison, the simulation scenarios examined referred to the same conditions referred to in [31]. The optimal values for the control parameters of the GA and PSO algorithms were determined based on suggestions found in literature, in conjunction with trial and error tests. To be more specific, different control parameter combinations were tested, within the ranges suggested in literature [37,38]. For each specific control parameter combination, 30 runs were conducted and the combination that produced the best value of the objective function on average during the 30 runs was finally chosen as the finest set of control parameters.

Specifically, the main control parameters for the GA developed are presented in Table 1.

**Table 1.** Genetic algorithm (GA) control parameters.

| Control Parameters | Functions/Values |
| --- | --- |
| Creation Function | 'CreationLinearFeasible' |
| Selection Function | 'SelectionStochUnif' |
| Crossover Function | 'CrossoverScattered' |
| Crossover Fraction | 0.5 |
| Mutation Function | 'MutationAdaptFeasible' |
| Population Size | [200, 600] |
| Function Tolerance | 0.01 (for 50 generations) |

The initial population on the chromosomes was created in a random way so that the bounds and the linear constrains were satisfied, provided that they existed. Later on, in order to choose the best parents, a stochastic method was used according to which the chromosomes of the population were placed as consecutive linear segments on a straight line so that the length of every segment was similar to its fitness. The algorithm searched on the line by making steps of the same size and in each step it chose the parent who corresponded to the linear segment on which it landed. The first step occurred as a random number, smaller than the size of the step. Afterwards, during the crossover process, a random binary vector of the same size as the chromosomes was created and in the points where

this vector took the value of "1", the genes of the first parent were selected, whereas when it took the value of "0", the genes of the second parent were selected. The selected genes were combined for the offspring-chromosome to occur. For the mutation process, a function that takes under consideration the bounds and the constraints, which could potentially occur, was chosen. The size of the population differs in respect to the complexity of the problem. Therefore as far as first two simpler problems were concerned, the size of the population was chosen to be equal to 200; while in the last two and most difficult cases the population was chosen equal to 600 chromosomes.

Similarly, the fundamental control parameters of the PSO algorithm were given the values that are presented in Table 2.

**Table 2.** Particle swarm optimization control parameters.

| Control Parameters | Functions/Values |
|---|---|
| Inertia Range (w) | [0.1, 1.1] |
| Self-Adjustment Weight $c_1$ | 1.49 |
| Social Adjustment Weight $c_2$ | 1.49 |
| Swarm Size | [200, 600] |
| Function Tolerance | 0.01 (for 20 iterations) |

The initial swarm was created by having the particles occupy random yet uniformly distributed positions in the area as long as these positions were within the boundaries. The coefficients of the social and cognitive part ($c_1$ and $c_2$) were chosen to have the same value, since the particles function in the most efficient way when these two constants co-exist harmonically; on the other hand the inertia weight was altered linearly so that its initial high value boosted global search capability (exploration of the area), whereas this value decreased during the execution of the algorithm so as to boost the partial search capability of the algorithm (exploitation). The size of the swarm differed in respect to the complexity of the problem. Therefore as far as the first two simpler problems were concerned, the size of the swarm was chosen to be equal to 200; while in the last two and most difficult cases the swarm size was chosen equal to 600 particles. In case studies 1 and 3 all sensor nodes were considered to have the equal sensing ranges, while in case studies 2 and 4 the sensor nodes were considered to have different sensing ranges.

In each case study the simulation test was executed 30 times and the mean value, the standard deviation of the area coverage, as well as the average time were calculated and are presented in corresponding tables along with the highest value of the area coverage which resulted from the execution of each case. The information concerning the number of evaluations of the fitness function, as well as the standard deviation of this value are also presented in the same table. Additionally, the final positions of the sensor nodes in the area, which were the optimal positions in terms of coverage maximization among these 30 results, are both found and depicted. Moreover, a graphical representation of the best value of the fitness function is provided in accordance with the iterations for one of the 30 executions of the algorithm. Referring to the first four case studies, it must be noted that the order of appearance in their presentation corresponds to a graded degree of difficulty starting from the easiest one to calculate. Also, referring to the last three case studies, the order of appearance in their presentation corresponds to that adopted in [31]. Finally, in all cases studies examined, the term ideal area coverage in the results presented refers to the sum of the area coverage of the available sensor nodes, without taking into consideration the overlaps that may occur among them.

In all of the case studies examined, a comparison among the algorithms investigated is performed. Specifically, in the first four case studies, the competing methodologies are the GA and PSO algorithms proposed, whereas in the last three case studies the comparison that takes place is among each one of the proposed algorithms and the algorithms presented in [30] and [31]. This comparison was carried out using t-test methodology, which allows statistical testing of a hypothesis applicable to a population. This methodology is based on the calculation of the p-value, which indicates how compatible the

data are, with a hypothesis known as the null hypothesis. As null hypothesis it is assumed that the two competing algorithms produce results with the same mean.

### 4.1. Case Study 1

The aim in the first case study was to maximize the coverage of a two-dimensional, squared area of $20 \times 20$. The number of the available sensors was equal to 35 and their sensing range was equal to 1.5. The size of the population in the GA algorithm was selected to be equal to 200 while the swarm size in the PSO algorithm was adjusted to 200.

The results of the simulations performed are synoptically presented in Table 3.

**Table 3.** Simulation results of case study 1.

| Parameter | GA | PSO |
|---|---|---|
| Mean Value | 61.17 | 60.92 |
| Standard Deviation | 0.28 | 0.46 |
| Average Time (sec) | 1.52 | 2.18 |
| Mean Function Evaluations | 17,200 | 27,400 |
| Standard Deviation Function Evaluations | 865 | 2300 |
| Best Fitness | 61.56 | 61.49 |
| p-value | 0.00 | |
| Ideal Area Coverage | 61.85 | |

Additionally, the optimal positions of the network nodes that are calculated with the use of the GA algorithm are depicted in Figure 3 while the best and mean fitness values for each generation are shown in Figure 4.

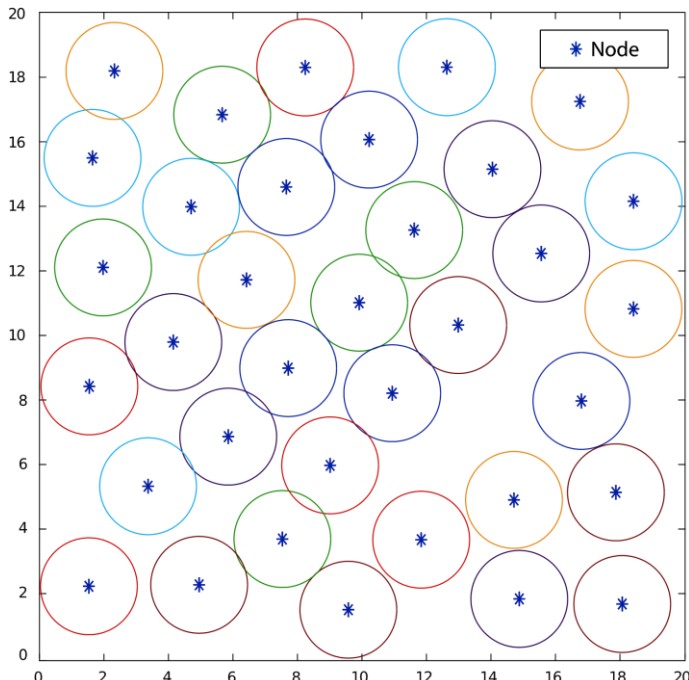

**Figure 3.** Optimal node position with the use of the GA algorithm in case study 1.

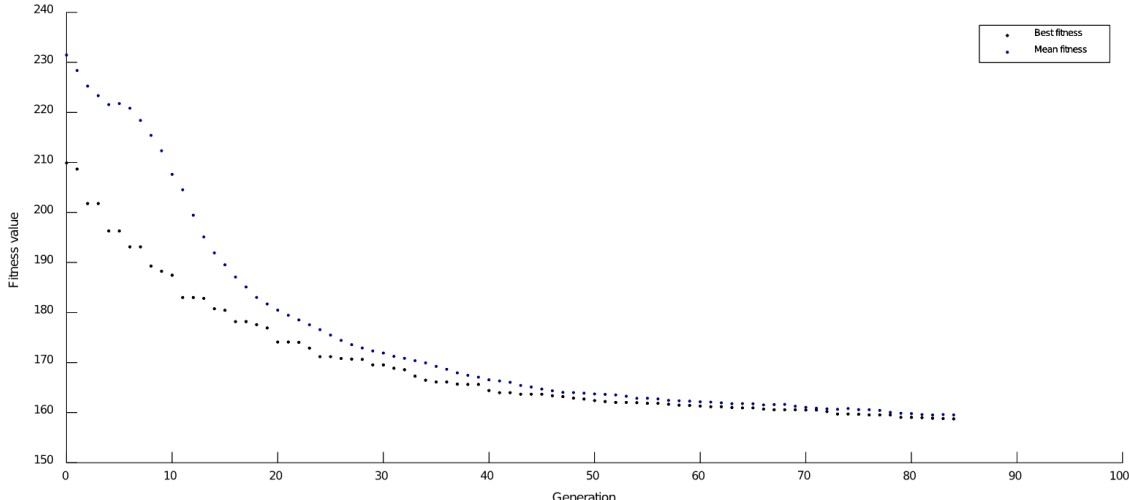

**Figure 4.** Best and mean fitness values for each generation of the GA algorithm in case study 1.

Similarly, the optimal positions of the network nodes calculated using the PSO algorithm are illustrated in Figure 5 while the best fitness values for each iteration are graphically presented in Figure 6.

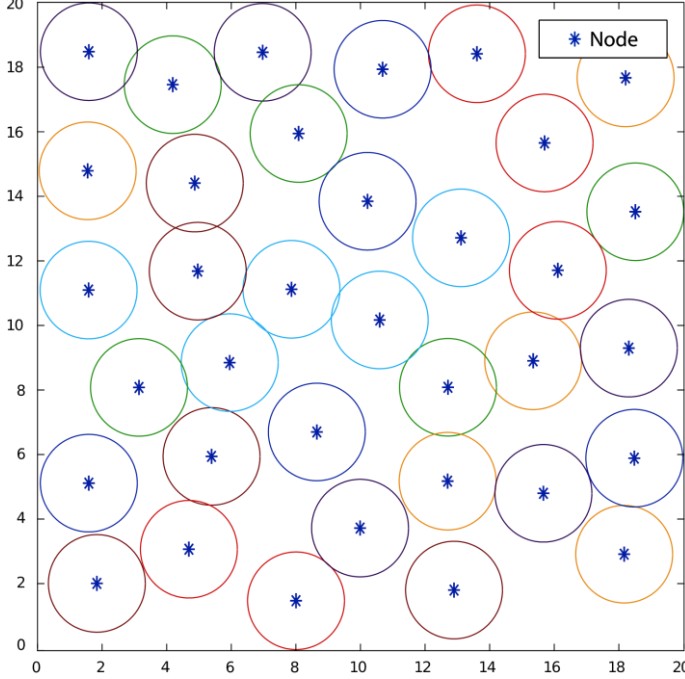

**Figure 5.** Optimal node positions with the use of the PSO algorithm in case study 1.

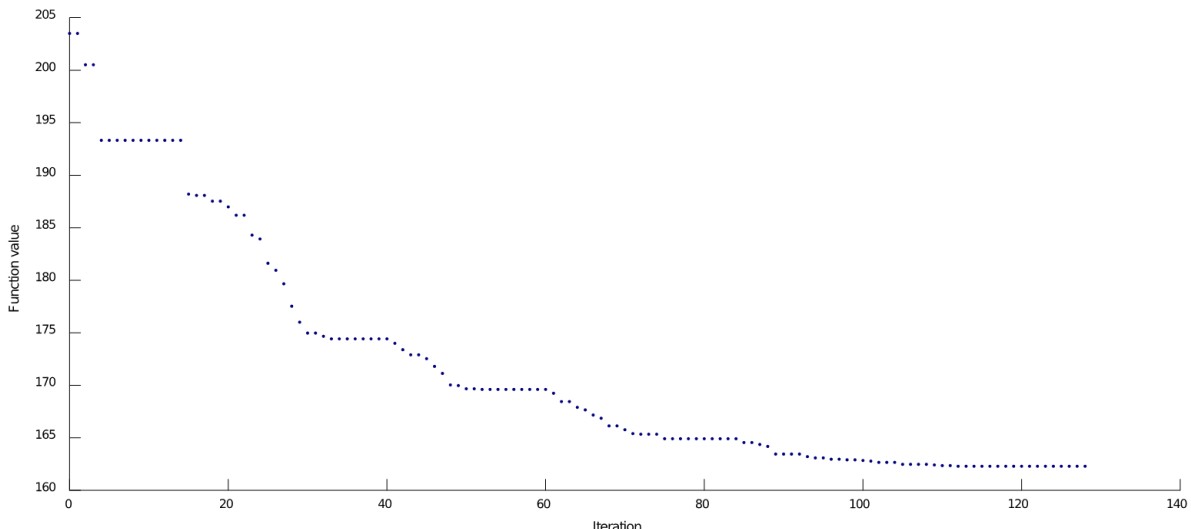

**Figure 6.** Best fitness value for each iteration of the PSO algorithm in case study 1.

*4.2. Case Study 2*

The aim in the second case study was to maximize the coverage of a two-dimensional, squared area of 20 × 20. The number of the available sensors was equal to 32 while their sensing range varied between 0.8 and 2.

More specifically, five sensors had a sensing range equal to 0.8, 20 sensors had a sensing range equal to 1.5, and the last seven had a sensing range equal to 2. The size of the population in the GA algorithm was selected to be equal to 200 while the swarm size in the PSO algorithm was adjusted to be equal to 200.

The results of the simulations performed in case study 2 are synoptically presented in Table 4.

**Table 4.** Simulation results of case study 2.

| Parameter | GA | PSO |
|---|---|---|
| Mean Value | 59.37 | 58.83 |
| Standard Deviation | 0.18 | 0.38 |
| Average Time (sec) | 1.43 | 2.18 |
| Mean Function Evaluations | 16,000 | 27,200 |
| Standard Deviation Function Evaluations | 693 | 3020 |
| Best Fitness | 59.69 | 59.32 |
| p-value | 0.00 | |
| Ideal Area Coverage | 59.85 | |

Additionally, the optimal positions of the network nodes calculated with the use of the GA algorithm are depicted in Figure 7 while the corresponding best and mean fitness values for each generation are shown in Figure 8.

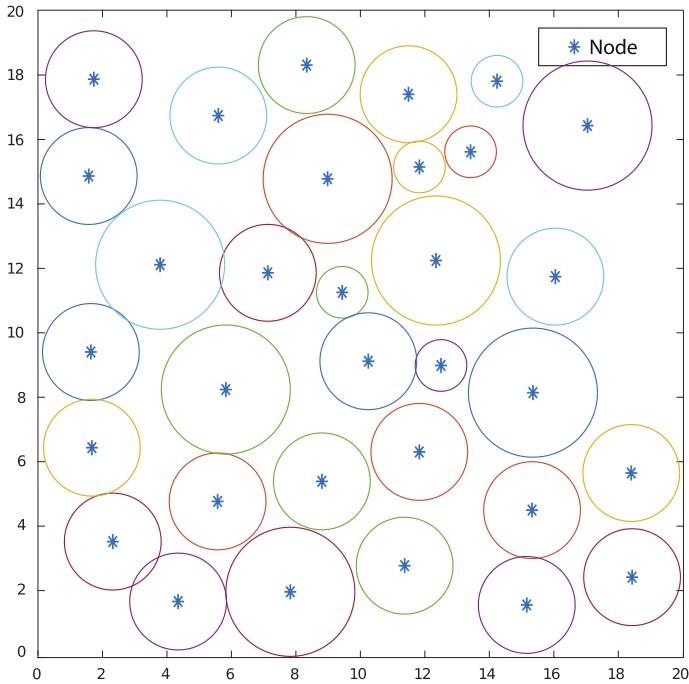

**Figure 7.** Optimal node positions with the use of the GA algorithm in case study 2.

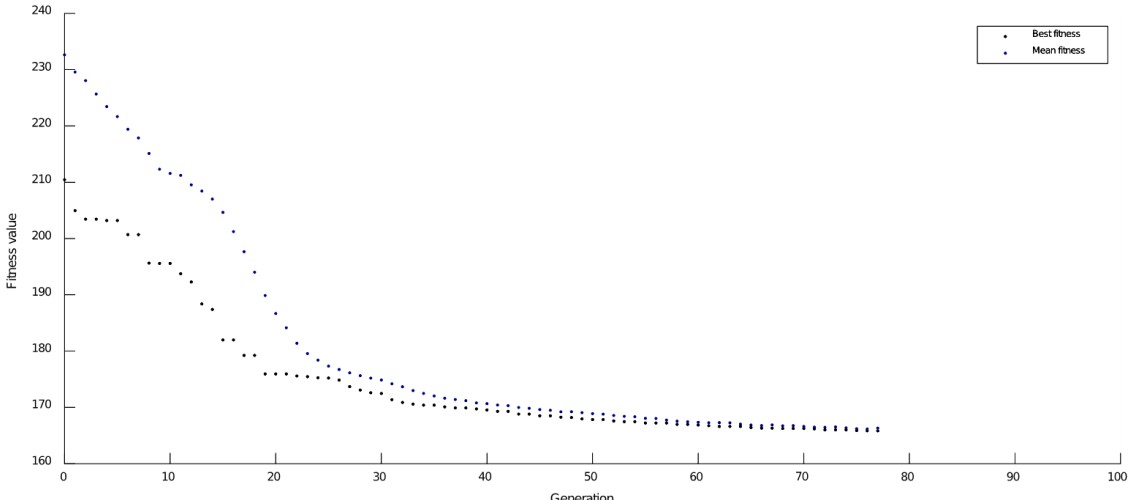

**Figure 8.** Best and mean fitness values for each generation of the GA algorithm in case study 2.

Similarly, the optimal positions of the network nodes calculated by using PSO algorithm are illustrated in Figure 9 while the corresponding best fitness values for each iteration are graphically presented in Figure 10.

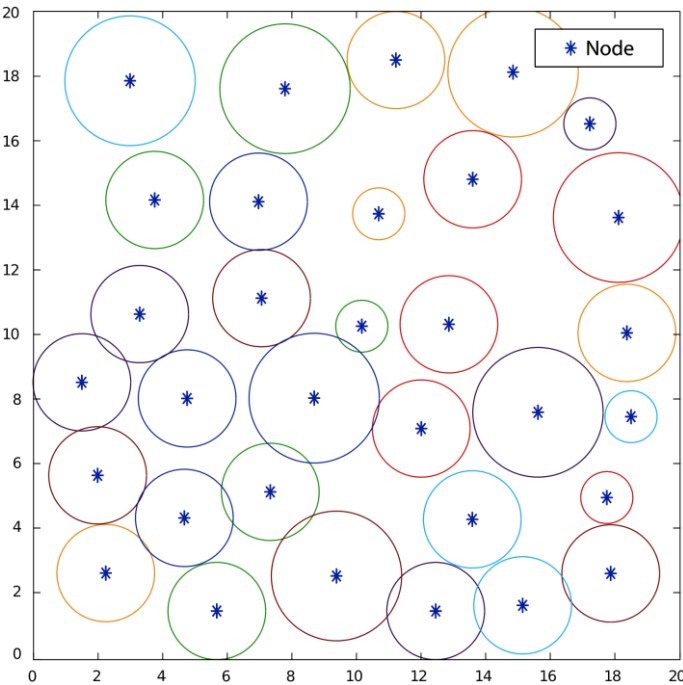

**Figure 9.** Optimal node positions with the use of the PSO algorithm in case study 2.

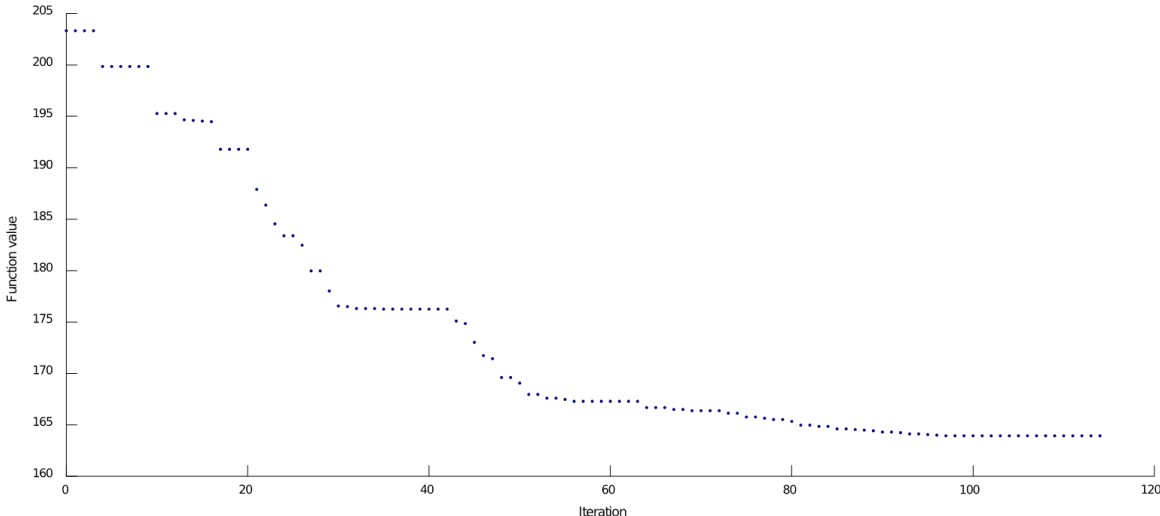

**Figure 10.** Best fitness value for each iteration of the PSO algorithm in case study 2.

### 4.3. Case Study 3

The aim in the third case study was to investigate the maximization of the coverage in a network located within a two-dimensional, squared area of $20 \times 20$ provided that k-coverage of six pre-defined spots was satisfied.

More specifically, the points $(5, 5)$, $(10, 5)$, $(15, 5)$, $(5, 15)$, $(10, 15)$, and $(15, 15)$ were considered to have three-coverage each one. The number of the available sensor nodes was equal to 45 while the sensing range of each of them was equal to 1.5. The size of the population used in the GA algorithm was selected to be equal to 600 while the swarm size in the PSO algorithm was adjusted to be equal to 600.

The results of the simulations performed are synoptically presented in Table 5.

**Table 5.** Simulation results of case study 3.

| Parameter | GA | PSO |
|---|---|---|
| Mean Value | 73.07 | 72.13 |
| Standard Deviation | 0.66 | 0.85 |
| Average Time (sec) | 7.6 | 5.15 |
| Mean Function Evaluations | 114,260 | 139,960 |
| Standard Deviation Function Evaluations | 12,300 | 12,700 |
| Best Fitness | 74.28 | 73.77 |
| p-value | 0.00 | |
| Ideal Area Coverage | 79.52 | |

Additionally, the optimal positions of nodes with the use of the GA algorithm are depicted in Figure 11 while the corresponding best and mean fitness values for each generation are shown in Figure 12.

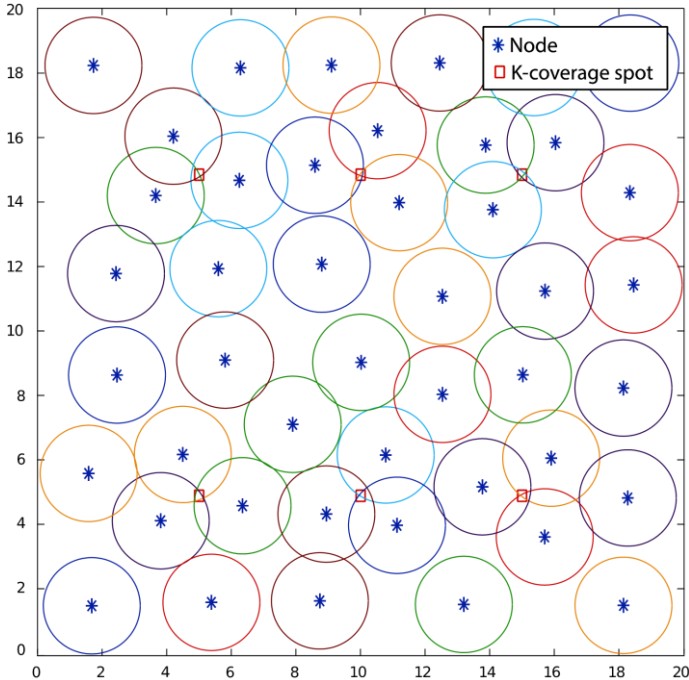

**Figure 11.** Optimal node positions with the use of the GA algorithm in case study 3.

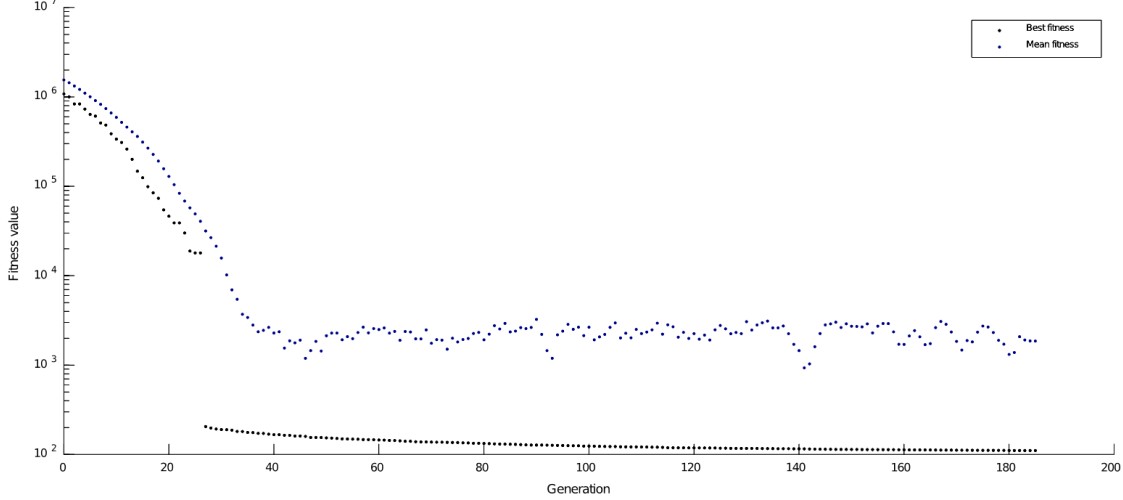

**Figure 12.** Best and mean fitness values for each generation of the GA algorithm in case study 3.

Similarly, the optimal positions of nodes calculated by using PSO algorithm are illustrated in Figure 13 while the corresponding best fitness values for each iteration are graphically presented in Figure 14.

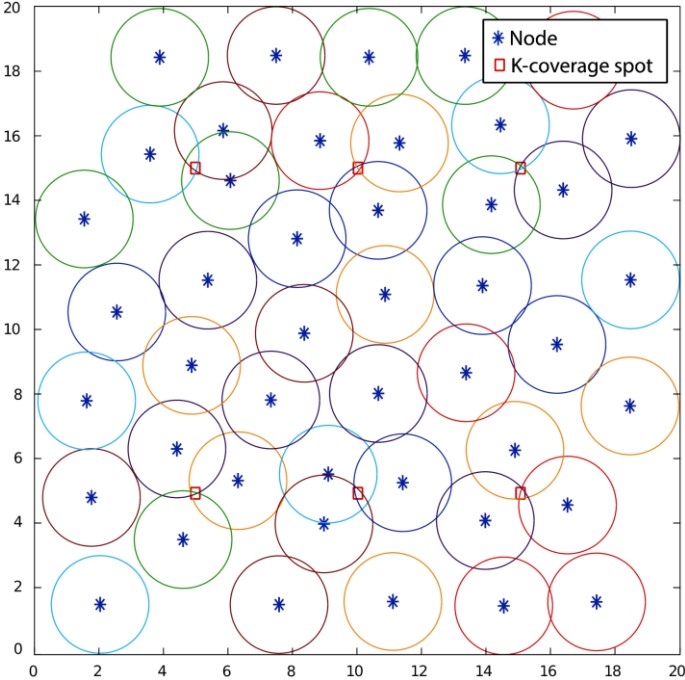

**Figure 13.** Optimal node positions with the use of the PSO algorithm in case study 3.

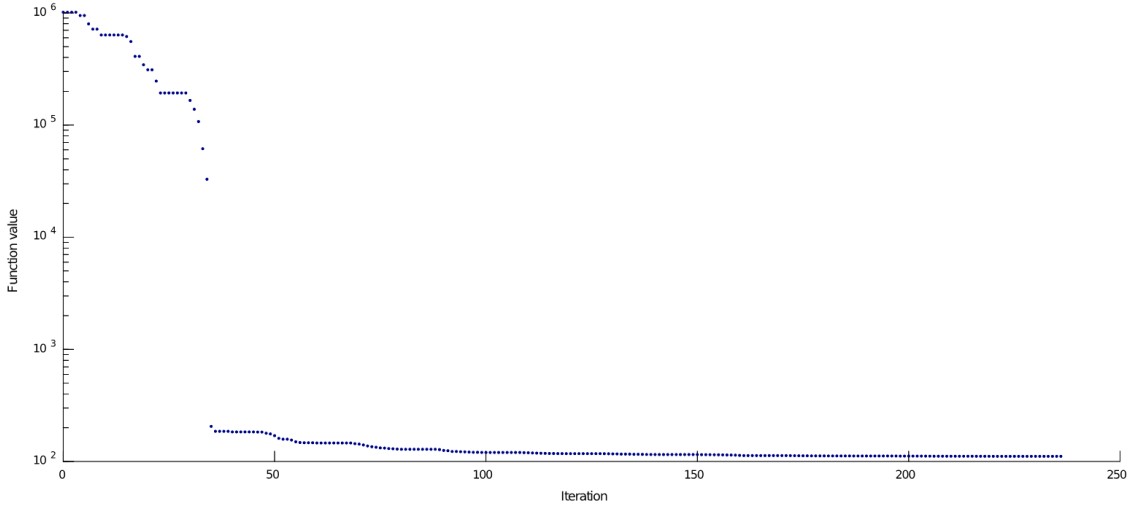

**Figure 14.** Best fitness value for each iteration of the PSO algorithm in case study 3.

### 4.4. Case Study 4

The aim in the fourth case study is to maximize the coverage of a two-dimensional, squared area of $20 \times 20$ provided that k-coverage of six pre-defined spots has been satisfied. More specifically, the points (5, 5), (10, 5), (15, 5), (5, 15), (10, 15), and (15, 15) are considered to have three-coverage each one. The number of the available sensors was equal to 45 while their sensing range varied between 1 and 2. More specifically, 18 sensors had a sensing range equal to 1, 20 sensors had a sensing range equal to 1.5, and the last seven had sensing ranges equal to 2. The size of the population in the GA algorithm was selected to be equal to 600 while the swarm size in the PSO algorithm was adjusted to 600.

The results of the simulations performed are synoptically presented in Table 6.

**Table 6.** Simulation results of case study 4.

| Parameter | GA | PSO |
|---|---|---|
| Mean Value | 67.39 | 69.89 |
| Standard Deviation | 0.45 | 1.15 |
| Average Time (sec) | 5.69 | 6.10 |
| Mean Function Evaluations | 96,340 | 136,180 |
| Standard Deviation Function Evaluations | $1.31 \times 10^4$ | $1.88 \times 10^4$ |
| Best Fitness | 68.24 | 71.64 |
| p-value | 0.00 | |
| Ideal Area Coverage | 71.47 | |

Additionally, the optimal positions of nodes with the use of the GA algorithm are depicted in Figure 15 while the corresponding best and mean fitness values for each generation are shown in Figure 16.

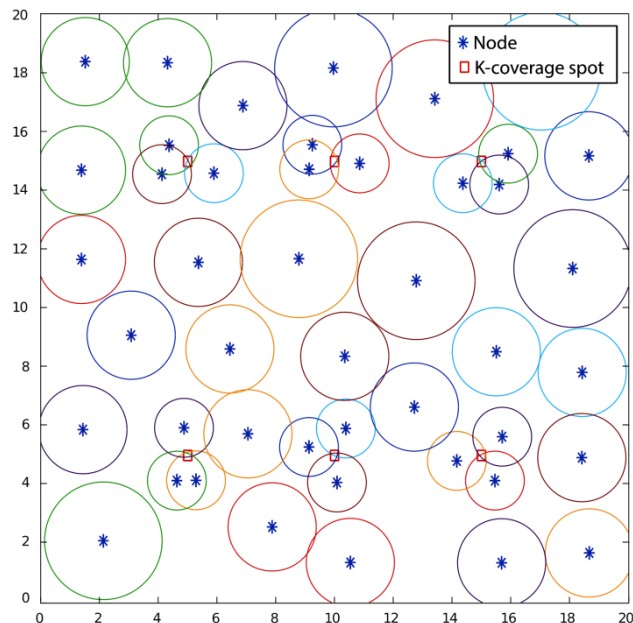

**Figure 15.** Optimal node positioning with the use of the GA algorithm in case study 4.

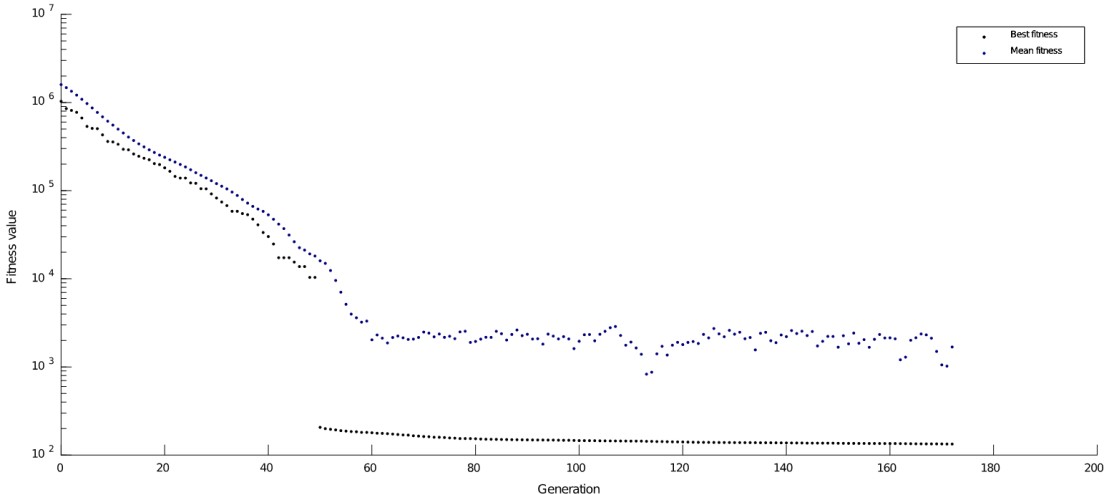

**Figure 16.** Best and mean fitness values for each generation of the GA algorithm in case study 4.

Similarly, the optimal positions of nodes calculated by using PSO algorithm are illustrated in Figure 17 while the corresponding best fitness values for each iteration are graphically presented in Figure 18.

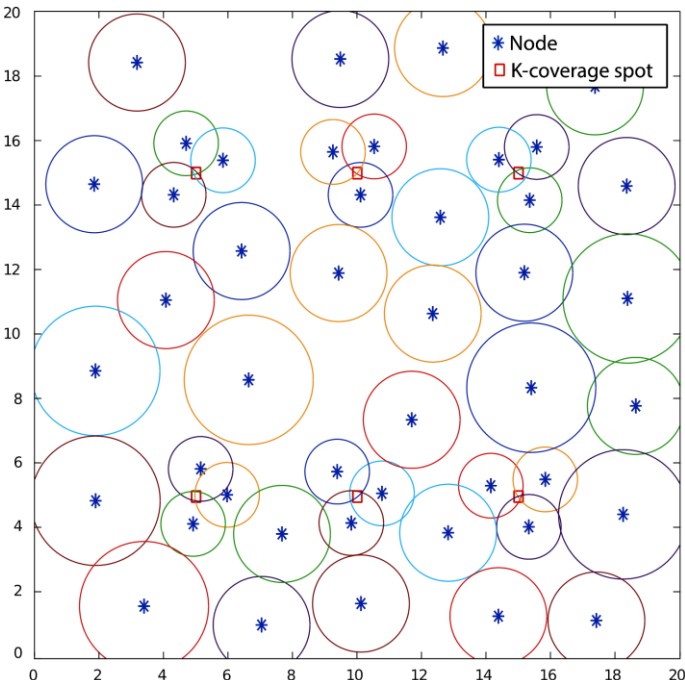

**Figure 17.** Optimal node positioning with the use of the PSO algorithm in case study 4.

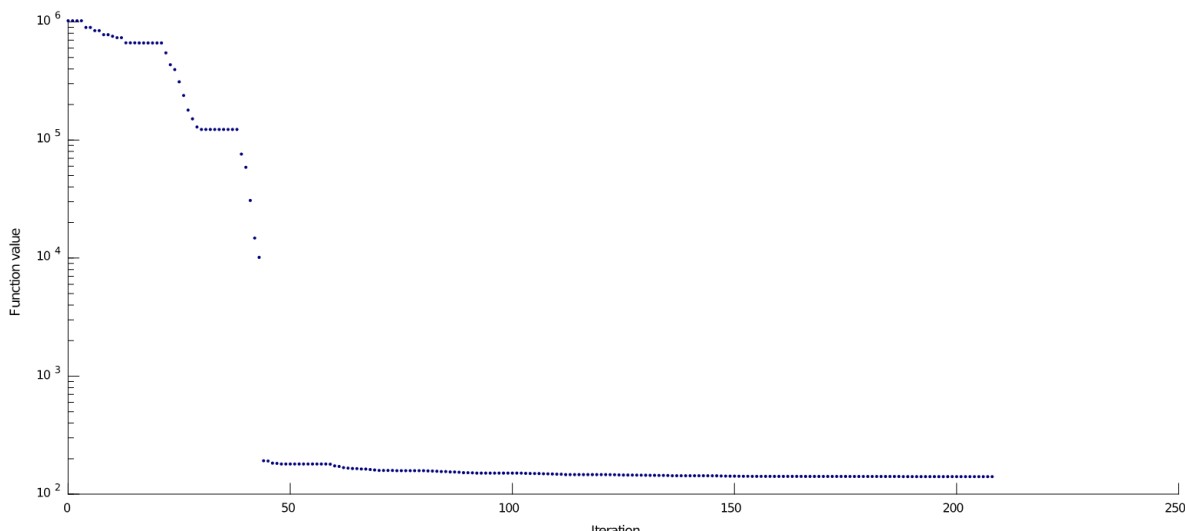

**Figure 18.** Best fitness value for each iteration of the PSO algorithm in case study 4.

*4.5. Case Study 5*

The aim in the fifth case study was to maximize the coverage of a two-dimensional, squared area of $50 \times 50$. The number of the available sensors was equal to 40 and their sensing range was equal to 5. The grid dimensions were $1 \times 1$, the size of the population in the GA algorithm was equal to 600, and the swarm size in the PSO algorithm was adjusted to 600. The results derived are synoptically presented in Table 7.

**Table 7.** Simulation results of case study 5.

| Values | PSO_GRID [30] | PSO_VORONOI [31] | GA | PSO |
|---|---|---|---|---|
| Mean Value | 91.74 | 89.83 | 96.40 | 95.53 |
| Standard Deviation | 0.01 | 0.01 | 0.59 | 0.66 |
| Average Time (sec) | 645.33 | 61.28 | 217.59 | 200.99 |
| p-value (GA) | 0.00 | 0.00 | - | 0.00 |
| p-value (PSO) | 0.00 | 0.00 | 0.00 | - |
| Ideal Area Coverage | | 100% | | |

### 4.6. Case Study 6

The aim in the sixth case study was to maximize the coverage of a two-dimensional, squared area of $50 \times 50$. The number of the available sensors was equal to 20 and their sensing range was equal to 5. The grid dimensions were $1 \times 1$, the size of the population in the GA algorithm was equal to 600, and the swarm size in the PSO algorithm was adjusted to 600. The results derived are synoptically presented in Table 8.

**Table 8.** Simulation results of case study 6.

| Values | PSO_GRID [30] | PSO_VORONOI [31] | GA | PSO |
|---|---|---|---|---|
| Mean Value | 61.07 | 59.24 | 62.50 | 62.38 |
| Standard Deviation | 0.00 | 0.00 | 0.23 | 0.18 |
| Average Time (sec) | 362.44 | 27.67 | 149.12 | 89.06 |
| p-value (GA) | 0.00 | 0.00 | - | 0.10 |
| p-value (PSO) | 0.00 | 0.00 | 0.10 | - |
| Ideal Area Coverage | | 62.83% | | |

### 4.7. Case Study 7

The aim in the seventh case study was to maximize the coverage of a two-dimensional, squared area of $30 \times 30$. The number of the available sensors was equal to 20 and their sensing range was equal to 5. The grid dimensions were $1 \times 1$, the size of the population in the GA algorithm was equal to 600, and the swarm size in the PSO algorithm was adjusted to 600. The results derived are synoptically presented in Table 9.

**Table 9.** Simulation results of case study 7.

| Values | PSO_GRID [30] | PSO_VORONOI [31] | GA | PSO |
|---|---|---|---|---|
| Mean Value | 97.89 | 97.96 | 99.76 | 99.55 |
| Standard Deviation | 0.02 | 0.01 | 0.10 | 0.17 |
| Average Time (sec) | 125.18 | 27.17 | 39.24 | 44.98 |
| p-value (GA) | 0.00 | 0.00 | - | 0.00 |
| p-value (PSO) | 0.00 | 0.00 | 0.00 | - |
| Ideal Area Coverage | | 100% | | |

## 5. Performance Evaluation

According to the outcome of the simulations presented above, both the algorithms present satisfactory results of coverage and k-coverage of the area, which are highly close to the ideal values for each case, within acceptable execution times.

A rather important factor which affects the quality of the solution as well as the computational execution time of the algorithm is the size of the population which is selected in the GA algorithm and the size of the swarm in the PSO algorithm. The choice of a large number of possible solutions, otherwise known as agents, exploring the search space increases the diversity, thus resulting in less iterations of the algorithm so as to converge to the optimal solution while at the same time the possibility of getting

trapped in local extremes decreases. Contrastively, the selection of a large population size concerning the GA algorithm as well as a large swarm size of particles as far as the PSO algorithm is concerned, increases to a great extent the complexity of the calculations. During the use of both of the methods, large sizes for the population and the swarm were chosen, since the need for finding the optimal coverage and k-coverage result was prioritized compared to the fast execution of the algorithms.

Additionally, it was observed that when the complexity of the problem increased, while maintaining the sizes of population in GA algorithm and swarm in PSO algorithm, the standard deviation in the simulation results was higher. Thus, while the difficulty of the problem increased, larger sizes of the populations were chosen. The fact that was noticeable, was the ability of the PSO algorithm to adapt and solve the most demanding problems, where the requirement of k-coverage was added, having a smaller swarm size compared to the GA algorithm where the respective population size was not sufficient to find a solution to the coverage problem. Nevertheless, for comparison purposes equal population sizes were chosen in all case studies examined.

As far as the mean value of the coverage calculation is concerned, the GA algorithm outperforms the PSO algorithm in all cases except for case study 4, where PSO produces a far superior result. The differences between the two algorithms were found to be statistically significant with a confidence level higher than 99% in all case studies, with the exception of case study 6, where the confidence level dropped to 90%.

As the complexity of the calculations increased due to the large number of potential solutions, the execution times of the algorithms were significantly increased as well. The execution times of the algorithms were also affected by the number of sensor nodes and not as much by the k-coverage points and the degree of coverage which they demand. Another factor which undoubtedly affects the execution time is the sampling step which, nevertheless, contributes to the improvement of the algorithm's results. What is considered as an issue of outmost importance is to define the optimal sampling step, since it was observed that after a certain point the further decrease of the sampling step led to a great increase of the time without the respective improvement of the area coverage. As far as the simpler cases are concerned, the execution of the GA algorithm demanded less time compared to the PSO algorithm; yet, as the difficulty of the experiments increased, the GA algorithm needed more time and a larger size of population so as to achieve a result comparable to the one that was calculated with the use of the PSO algorithm.

The results derived from a group of 30 executions of the algorithm for each case and in each one of them, the standard deviation was rather small, depending on the complexity of the problem. Based on these small standard deviation values, it was concluded that the algorithm's results, having used both methods, present a high repeatability.

Furthermore, the results derived were close enough to the ideal coverage percentage value. Even though some cases where the problem needed k-coverage of certain points and the existence of overlaps within the sensor nodes—which is not calculated in the ideal area coverage—were inevitable, the results of the optimal solution were very close to the ideal optimal values. It is, thus, observed that through the use of the GA as well as the PSO algorithm a high degree of accuracy concerning the results can be achieved.

The examination of the last three case studies showed that both the proposed algorithms are more advantageous compared to the grid-based algorithm proposed in [30] as far as the average time is concerned, but not as good as the Voronoi-based algorithm proposed in [31]. This is reasonable because the construction of the Voronoi diagram is a less complicated approach than the sampling methodology. Nevertheless, both of the algorithms proposed were found to be superior compared to the older two approaches in all of the case studies examined in terms of the area coverage ratio. The superiority of the proposed algorithms over the algorithms proposed in [30] and [31] is statistically significant with a confidence level higher than 99%, since the p-value is always very close to 0.

## 6. Conclusions and Future Work

The optimization of the area coverage in WSNs is the subject of the research work presented in this article. Specifically, two computational intelligence algorithms were designed in order to achieve the maximization of area coverage and area k-coverage. Both the genetic algorithm and the particle swarm optimization algorithm developed were analytically compared against each other in the various scenarios examined, using statistical testing.

In all case studies examined, the efficacy of the two algorithms developed was found to be very close to the ideal in terms of the achieved coverage percentage. More precisely, it was found that in the simpler case studies examined, the execution of the GA required less execution time compared to the PSO algorithm. Yet, as the scale of the experiments increased, the genetic algorithm needed more time and a larger size of population in order to achieve results as well as those accomplished by using the PSO algorithm. Similarly, in the simpler case scenarios examined, the GA achieved a minimally better result in terms of mean value of coverage calculations compared to the PSO algorithm.

In the cases where the k-coverage was pursued, the difference in the mean value of coverage calculated when using both the algorithms was reduced and finally in the most complex case scenario, the PSO algorithm was found to be better. Additionally, the GA and the PSO algorithms developed were analytically compared against two well-known coverage maximization methods, i.e. a grid based PSO and a Voronoi-based PSO algorithm, using statistical testing. Both the algorithms proposed in this article were found to provide higher coverage percentages than the other algorithms in comparison, with execution times lower than the former of them and higher than the latter one.

Despite the promising results derived from this work, there are a lot of efforts that need to be made in order to achieve even better performance enhancement of WSN performance. The reason for this statement is that there are many problems that generally obstruct the operation of WSNs and in particular affect either directly or indirectly the coverage optimization. For instance, restrictions of energy in WSNs have a detrimental influence on coverage maximization. This is because the energy insufficiency of sensor nodes limits their operational life [39], and whenever the residual energy of a sensor node is exhausted, this node becomes inactive thus causing coverage holes. Consequently, the use of schemes aiming at the achievement of power control [40], data aggregation [41,42], energy efficiency [43–45], energy balancing [46–48], data compression and restoration [49–51], congestion avoidance, and congestion control [52–56] should be also considered. Likewise, the preservation of network connectivity is an issue of crucial importance for WSNs in order to be operational and thus has to be pursued [57–60]. Yet, the deployment of sensor nodes at the network field of a WSN affects not only connectivity but also coverage. This is because in order to ensure connectivity, the sensor nodes need to be placed close enough to each other so that they remain within their communication range. On the other hand, in order to maximize the coverage, the sensor nodes need to be scattered around the area so as to prevent forming coverage holes.

For the aforementioned reasons, in order to achieve optimal results, coverage maximization should be pursued in combination with the attainment of energy conservation, connectivity preservation, and other performance metrics through the application of proper multiobjective schemes [61–67]. This is the next challenge for the authors of this article in terms of future research work.

**Author Contributions:** All authors have contributed equally to conceptualization, methodology, formal analysis, investigation, validation, resources, original draft preparation, and review and editing of this research article; I.P. and K.T. were responsible for software and data curation; A.A. and D.K. carried out the overall supervision and project administration. All authors have read and agreed to the published version of the manuscript.

**Funding:** This research received no external funding.

**Conflicts of Interest:** The authors declare no conflict of interest.

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
