# Peer review of "Coverage and k-Coverage Optimization in Wireless Sensor Networks Using Computational Intelligence Methods: A Comparative Study"

_electronics, doi:10.3390/electronics9040675_

Round 1
Reviewer 1 Report
In this paper, authors proposed two schemes based on computational intelligence for coverage and k-coverage optimization in WSN.
The main strength of the proposed schemes is to adapt genetic algorithm and particle swarm optimization algorithm to coverage and k-coverage optimization in WSN. Moreover, authors demonstrate that the proposed schemes outperform the related work through simulation.
However, the paper still needs to be enhanced in terms of the following points.
First, the paper does not seem to be read well. The paper substantially needs to be polished for better presentation.
Second, the proposed schemes should be thoroughly compared to the related work. Although the comparison results are presented in the paper, they are not enough to demonstrate that the proposed schemes outperform substantial number of related work. This is because a lot of related work for performance comparison exist in the field of coverage and k-coverage optimization in WSN .
Author Response
In this paper, authors proposed two schemes based on computational intelligence for coverage and k-coverage optimization in WSN.
The main strength of the proposed schemes is to adapt genetic algorithm and particle swarm optimization algorithm to coverage and k-coverage optimization in WSN. Moreover, authors demonstrate that the proposed schemes outperform the related work through simulation.
However, the paper still needs to be enhanced in terms of the following points.
Comment 1:
First, the paper does not seem to be read well. The paper substantially needs to be polished for better presentation.
Respond to Comment 1:
The authors thank the reviewer for pointing this out. Following this suggestion, the article has been revised. Specifically, explanatory information has been added in the Abstract, Section 5, Section 6 and Section 7 in order to enhance the comprehension of context. Furthermore, the article was checked once again in order to find and eliminate grammar and syntactic mistakes existing.
Comment 2:
Second, the proposed schemes should be thoroughly compared to the related work. Although the comparison results are presented in the paper, they are not enough to demonstrate that the proposed schemes outperform substantial number of related work. This is because a lot of related work for performance comparison exist in the field of coverage and k-coverage optimization in WSN.
Respond to Comment 2:
It is true, as the reviewer mentions, that the optimization of coverage and k-coverage in WSNs is a very important issue that keeps on attracting research interest. The aim of this article is to pursue the maximization of area coverage and area k-coverage, by using two different computational intelligence algorithms, i.e. PSO and GA, and provide a comparison study between these two approaches. Two different well-known algorithms were also used for comparison purposes. Furthermore, in order to provide a more thorough comparison to the related work, a statistical testing procedure was followed in this revised version of the manuscript.
Reviewer 2 Report
This paper presents two metaheuristics for coverage and k-coverage optimization in wireless sensor networks. I have two comments for the experiments section. 1. The parameters of GA and PSO should be fine-tuned by a design-of-experiments approach in order to identify the best values. 2. The comparison of algorithms should rely on a statistical test to reveal whether the differences between the algorithms are significant enough.Author Response
This paper presents two metaheuristics for coverage and k-coverage optimization in wireless sensor networks. I have two comments for the experiments section.
Comment 1:
The parameters of GA and PSO should be fine-tuned by a design-of-experiments approach in order to identify the best values.
Respond to Comment 1:
The optimal values for the control parameters of the GA and PSO algorithms were determined based on suggestions found in literature, in conjunction with trial and error tests. To be more specific, different control parameter combinations were tested, within the ranges suggested in literature. For each specific control parameter combination, 30 runs were conducted and the combination that produced the best value of the objective function on the average of the 30 runs was finally chosen as the finest set of control parameters. This procedure had already been followed in the first original version of the manuscript, but the authors omitted to report it. For this reason, the authors would like to thank the Reviewer for pointing out this omission. In this revised version of the manuscript, relative explanatory text was added in page 7, along with 2 new bibliographic references ([37] and [38]).
Comment 2:
The comparison of algorithms should rely on a statistical test to reveal whether the differences between the algorithms are significant enough.
Respond to Comment 2:
Incorporating this valuable recommendation of the reviewer, the authors applied a statistical testing procedure based on t-test, which provided the necessary support to the experimental work performed. Extensive changes were applied in the experimental section, in order to present and discuss the results of statistical testing.
Round 2
Reviewer 2 Report
The revision is acceptable.